# A Virtual Knowledge Graph for Enabling Defect Traceability and Customer Service Analytics

Nico Wilhelm[1(✉)], Diego Collarana[2] ⓘD, Jens Lehmann[2]

[1] ZF Friedrichshafen AG
`nico.wilhelm@zf.com`
[2] Fraunhofer IAIS and University of Bonn, Germany
`{diego.collarana.vargas|jens.lehmann}@iais.fraunhofer.de`

**Abstract.** In this paper, we showcase the implementation of a semantic information model and a virtual knowledge graph at ZF Friedrichshafen AG company, with two main goals in mind: 1) integration of heterogeneous data sources following a pay-as-you-go approach; and the 2) combination core domain concepts from ZF's production line with meta-data of its internal data sources. We employ the developed semantic information model in two use cases, defect traceability and customer service, demonstrating and discussing the benefits and opportunities provided by following an agile semantic virtual integration approach.

## 1 Introduction

ZF Friedrichshafen AG, with a 105 years history, is a world-leading supplier of mobility systems for passenger cars, commercial vehicles, and industrial technology. The division Electrified Powertrain Technology (E-division) provides various electrified and conventional mobility applications within the passenger cars segment for decades. Therefore, diverse and complex data ecosystems have emerged. However, meta and context data is rarely formally documented. Moreover, if expressed at all, the meaning of business entities is not commonly shared within all domains (e.g., Production, Quality, among others). The E-division partly achieves data integration through platforms and data lakes, yet business entities' interlinking is missing. Hence, ZF requires new methods for holistic data enablement, defining common data and semantic standards, describing metadata to virtually explore all corresponding business data.

The fundamental challenge is to keep as much data "as it is" in the sources without expensive migration projects. A core requirement is creating an enterprise-wide model that defines the main business entities, e.g., "Product," "Machine," from a syntactic, but more importantly, from a semantic perspective creating a shared division-wide conceptualization of the domain. Furthermore, this semantic model must uniquely identify business concepts across all sources in the same way. A virtual semantic knowledge graph provides all elements to address those challenges. ZF and Fraunhofer IAIS partnered to implement such a semantic model to integrate ZF's data sources virtually. We prove the semantic model's value with two use cases following a "T-Shape" query principle. First, defect tractability for identifying domain-specific details. Second a warranty return case connecting several sources to support a broad information picture.

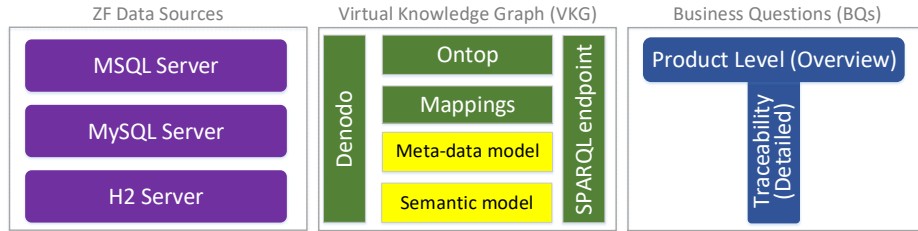

Fig. 1: Architecture

## 2    Virtual Knowledge Graph Integration Approach

Over the years, ZF has developed several data sources mainly composed of relational databases. Thus, different DBMS store heterogeneous but complementary knowledge about ZF's business entities and processes. Although useful for data analytics, these data sources generate different challenges related to knowledge integration and querying. By providing a Virtual Knowledge Graph (VKG) [5], we pursuit two goals. First, minimize the time required to answer new business queries (agile integration approach). Second, analyze the quality of data with a global approach (semantic data quality index). Following a pay-as-you-go integration approach [3], we have defined an innovative semantic layer for accessing ZF's existing data sources.

Figure 1 depicts the main elements of the semantic layer. The integration process starts by defining a set of **Business Questions (BQs)** that the VKG needs to answer. Following a T-Shape query principle, we firstly define overview questions. For example, in the defect traceability scenario, we start with a BQ like: "Show me quality, field, and plant data about my product." This BQ is the essential question starting quality-related product root cause analysis. Then, we add more triple patterns to answer detailed questions. For example, issue effects are evaluated through traceability BQs, e.g., "Show me all materials, tools, machines related to faulty products," thus, supporting domain production-specific drill-down questions. Once the BQs are defined, we extend the **Virtual Knowledge Graph** by updating entities and relationships in the semantic model required to answer those BQs. We use VoCol [1] to extend the semantic data model collaboratively. An innovation here is the combination of two data contexts to facility provenance of the data, i.e., two models encode both the domain and metadata of the sources linked together at mapping time. Then, we transform BQs into SPARQL queries validating whether all required entities are mapped from ZF's data sources. When necessary, we create new mappings using the R2RML [4] standard. Ontop [6] and Denodo [2] are our engines to transform the SPARQL queries into SQL queries that the original data sources understand. **ZF Data Sources** layer is composed of different DBMS, including Microsoft SQL Server, MYSQL, and H2, and they remain intact.

## 3    Lessons learned and benefits

During the development of the project, ZF gained valuable *lessons*. ZF applied a new way to integrate data vertically and horizontally across different data sources semanti-

cally. We validated the approach of modeling the domain of discourse based on Business Questions. ZF got an innovative methodology to rethink the structure of current data sources regarding their ability and connectivity. We develop basics design guidelines to disseminate this new methodology company-wide. The most evident *benefits* are the following. ZF enhances data exploration, i.e., domains can be easily explained and discovered. The designed ontology serves as a data and role model for future applications across different ZF's units. ZF can now provide domain language standardization, and the definitions can be experienced. So far, standards were just part of a document on a random share drive. ZF can apply data quality methods over the semantic layer where data from different sources is connected.

## 4   Conclusion and Future Lines of Work

The virtual knowledge graph data integration and access approach are under evaluation by different units at ZF. So far, the project has revealed business potentials by providing an intuitive data retrieval process based on semantics and metadata guidance. Modeling business entities semantically, linking them to their metadata, and defining fundamental properties provide a closed-loop view on data that was not possible before. ZF is now extending this approach to make it company-wide applicable.

**Acknowledgements:** We acknowledge the support of the EU H2020 Projects Opertus Mundi (GA 870228) and LAMBDA (GA 809965), and the Federal Ministry for Economic Affairs and Energy (BMWi) project SPEAKER (FKZ 01MK20011A).

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
