# OpenReview forum: "A Virtual Knowledge Graph for Enabling Defect Traceability and Customer Service Analytics"
_eswc-conferences.org/ESWC/2021/Conference/Industry_Track — ESWC 2021 Industry_

### Official Review · ~Yushan_Liu1 · 2021-04-15
**Interesting work on data integration using knowledge graphs**

**Rating:** 8
**Confidence:** 4

**Review:**

In the paper, a virtual knowledge graph is described, which aims at consistently integrating heterogeneous information across the whole company for different tasks like defect traceability or customer service analytics. The paper is well written and addresses a relevant problem.

Strengths:
- Clearly written and well structured
- Comprehensive description of the challenge and motivation
- A virtual knowledge graph combining information from different sources and division is important for efficiently querying the databases and getting a better understanding of the data
- Good overview of the semantic layer

Weaknesses:
- Detailed business questions need to be manually defined; expert domain knowledge necessary
- No evaluation of the approach yet available

---

### Official Review · ~Nelia_Lasierra1 · 2021-04-16
**Experience using knowledge graphs for data integration in industry**

**Rating:** 9
**Confidence:** 4

**Review:**

This paper presents the implementation of a semantic model and a virtual graph evaluated for two use cases at ZF Friedrichshafen AG company. This solution aimed at integrating heterogeneous data sources and combination of domain concepts with internal metadata-data.

The paper is very well written and clear. The authors describe the use cases and point out the value it provides to ZF. The maturity of the solution is not very clear as the authors indicate it is under evaluation by different units at ZF. The authors point to some challenges in the industrial setting for the implementation of the approach and provide information of how those were addressed. Learning more about the challenges (or not) of its adoption and also about the end users acceptability would be valuable for the audience. I believe it would be of great interest for the audience of the Industry track to learn more about this work at the event.

---

### Official Review · ~Josiane_Xavier_Parreira1 · 2021-04-16
**Good example of the use of SW. Concrete benefits are still under analysis**

**Rating:** 6
**Confidence:** 4

**Review:**

The paper describes a use case for enabling integration of data coming from different divisions within a company. This particular example comes from ZF Friedrichshafen AG.

Both motivation and goals are clear and well defined. The approach uses a combination of different tools to support data modelling, data mapping, and query answering. The solution opts for a Virtual KG approach to keep in the original sources. The solution is currently under evaluation within the company.

The paper lists the motivations for the solution, and in particular for the choice of a VKG. As evaluation is still underway it is currently not possible to properly assess the benefits of the approach, for instance, w.r.t. time required to answer new queries and data quality. Example queries were given, but it would be helpful to include how these queries are currently being answer, i.e. how many sources need to be queries, how long does it take, etc.

To summarize, a good example of the use of SW but the concrete benefits are still under analysis.